# From Crisis to Opportunity: A Qualitative Study on Rehabilitation Therapists’ Experiences and Post-Pandemic Perspectives

**DOI:** 10.3390/healthcare12101050

**Published:** 2024-05-20

**Authors:** Marianne Saragosa, Farwa Goraya, Frances Serrano, Behdin Nowrouzi-Kia, Sara Guilcher, Yasmin Abdul Aziz, Basem Gohar

**Affiliations:** 1Department of Population Medicine, University of Guelph, Guelph, ON N1G 2W1, Canada; fgoraya@uoguelph.ca (F.G.); fserrano@uoguelph.ca (F.S.); bgohar@uoguelph.ca (B.G.); 2Department of Psychology, University of Guelph, Guelph, ON N1G 2W1, Canada; 3Department of Psychology, Laurentian University, Sudbury, ON P3E 2C6, Canada; 4Department of Occupational Science & Occupational Therapy, University of Toronto, Toronto, ON M5G 1V7, Canada; behdin.nowrouzi.kia@utoronto.ca; 5Temerty Faculty of Medicine, University of Toronto, Toronto, ON M5S 1A8, Canada; 6Centre for Research in Occupational Safety & Health, Laurentian University, Sudbury, ON P3E 2C6, Canada; 7Department of Physical Therapy, University of Toronto, Toronto, ON M5G 1V7, Canada; sara.guilcher@utoronto.ca; 8Leslie Dan Faculty of Pharmacy, University of Toronto, Toronto, ON M5S 3M2, Canada; yasmin.abdulaziz@utoronto.ca

**Keywords:** occupational therapists, physiotherapist, COVID-19, stress, healthcare provider, lessons learned

## Abstract

Rehabilitation therapists (RTs) have developed substantial mental health problems since the pandemic. Our study aimed to understand the experience of COVID-19 on occupational therapists and physiotherapists practicing in Canada, how the pandemic may have affected care delivery, and to identify new learnings articulated by RTs. A qualitative descriptive study design guided data collection through one-on-one interviews, dyadic interviews, and focus groups. We recruited active RTs across Canada, advertising on professional practice networks and social media platforms and using snowball sampling. Forty-nine RTs representing seven Canadian provinces participated. The four overarching themes developed using thematic analysis were (1) navigating uncertainty along with ever-changing practices, policies, and attitudes, (2) morphing roles within a constrained system, (3) witnessing patients suffering and experiencing moral distress, and (4) recognizing the personal toll of the pandemic on self and others, as well as lessons learned. Our study demonstrated that many RTs suffered moral distress, poor mental health, and some from challenging financial situations, especially those in the private sector. They also expressed a resilient attitude in response to these stressors. Implications in the future include identifying promising communication strategies that could act as protective factors, addressing workforce constraints and diminishing resources through innovative models of care.

## 1. Introduction

The COVID-19 pandemic caused an unprecedented global crisis involving disrupted health systems, gross economic and social burdens, and high mortality rates affecting the most vulnerable [1]. Early evidence from the pandemic has also shown the detrimental long-term effects on persons with post-COVID-19 symptoms [2]. Almost half (45%) of COVID-19 survivors experience a range of unresolved symptoms for months post-infection [3], impacting social and family relationships, employment, and quality of life [4]. As such, there are expected ongoing healthcare system strains and broader implications for public health [5]. During the initial waves, the rapid transmission of the virus challenged healthcare systems and caused healthcare professionals to struggle with clinical and nonclinical stressors [6]. Later, the uptake of and notable hesitancy regarding the COVID-19 vaccines was an additional source of stress among healthcare workers [7]. Unsurprisingly, high anxiety, fear, hopelessness and helplessness were prevalent among healthcare providers [8], although particular focus has been placed on physicians and nurses [9,10,11], leaving other rehabilitation healthcare professionals understudied [12,13].

Occupational therapists and physiotherapists (hereinafter referred to as rehabilitation therapists or RTs) play an essential role in improving the lives of individuals, groups and communities by helping to maintain health for people of all ages and enabling participation in meaningful activities [14,15]. They work in various healthcare contexts, providing direct patient care, education, and advocacy [16,17]. The COVID-19 pandemic significantly changed how RTs delivered interventions [18] and, in many cases, necessitated the rapid deployment of telehealth models [19], among other changes. Even before the pandemic, RTs were experiencing job stress and burnout [20,21]. Burnout is associated with drug and alcohol abuse [22] and is associated with suicidality [23], threatening the well-being of health professionals and the quality and safety of healthcare systems [24]. RTs’ work requires close contact with patients’ physical and psychological pain, often triggering emotional responses in themselves [25].

Notably, RTs have developed substantial mental health problems since the pandemic, evidenced by high levels of anxiety and depression compared to the average rate in the general public [26]. Anxiety and especially burnout are strong predictors of early retirement [27] and turnover intention [28], while resilience and perceived support from the organization could act as protective factors against burnout and turnover [27]. Despite this awareness of the adverse psychological outcomes of the pandemic and related impact on staffing, we know little about the impact of COVID-19 on RTs and their perceived stressors during the pandemic [29].

Therefore, our study aimed to understand the experiences of COVID-19 among understudied rehabilitation healthcare professionals, occupational therapists and physiotherapists practicing in Canada, how the pandemic may have affected care delivery, and identify new learnings articulated by RTs from the pandemic.

## 2. Materials and Methods

### 2.1. Study Design

This study used a qualitative descriptive study design [30] to explore RTs’ lived experiences during the pandemic. This design is suitable for the study given its exploratory and flexible nature [31] and ability to provide a straightforward description of events using the collected language [32]. This paper is also reported according to the consolidated criteria for reporting qualitative research (COREQ) guidance [33].

### 2.2. Participants and Setting

This study was conducted across Canada, and registered occupational therapists and physiotherapists with practice experience were recruited during the pandemic. This study’s authors used purposeful sampling strategy and a multipronged recruitment strategy. First, partnering organizations, like professional associations and regulatory bodies, disseminated the approved study material to their members through their internal networks (i.e., listservs). Second, through snowball sampling, existing participants shared the recruitment invitations with other RTs [34]. Last, we leveraged social media platforms (i.e., LinkedIn and Twitter) by sharing the recruitment flyer and inviting those interested in contacting the senior author (BG). As an incentive, each participant received a CAD 50.00 gift card. Given the increasing concerns related to fraudulent activities in online recruitment, notably when financial incentives are involved, we conducted thorough checks [35]. This included verifying participants’ identities by cross-referencing their names with publicly available records and confirming their job roles at the outset of each interview.

### 2.3. Data Collection

The senior author (BG), a clinical psychologist and public health researcher with expertise in mixed methods research, conducted all the interviews virtually over MS Teams with participants. A mix of one-on-one interviews, dyadic interviews, and focus groups were employed for this study. The method relied on participant availability and preference. One focus group took place in November 2022. Due to the holidays and scheduling preferences, recruitment resumed in 2023 between March and July 2023. The research team (FS, FG) audited and transcribed all the interviews verbatim. The semi-structured interview guide consisted of a short list of open-ended questions that followed a chronological approach that started with demographic questions, their experience working during COVID-19, professional and personal impact, and lessons learned from the pandemic through an RT lens. The iterative nature of data collection and analysis resulted in changes to the guiding question as the study progressed [36], including additional probes that explored emerging topic areas [37]. Ethics approval was obtained from the University of Guelph’s research ethics board (REB#22-03-001, approval date: 4 May 2022). All participants in this study provided written consent by email before the interview.

### 2.4. Data Analysis

This study used thematic analysis to analyze the text data. The inductive analytical approach followed the codebook approach described by Braun and Clarke [38]. Once transcribed, two coders independently and inductively coded the first five transcripts using ‘open codes’ (MS, FS) [39]. The objective during this stage was to code as many meaningful segments as possible [39]. Collectively, the coders discussed the identified codes, collated them, and co-developed a codebook that was updated with new codes and applied deductively throughout the analysis process. After the initial coding, all the codes were clustered and grouped by themes. When coding the remaining transcripts, modifications were made to the codebook—to the code names. Elements of the broadness of the study aim, narrative quality, cross-case analysis, and denseness of the sample specificity determined information power [40]. All the analyses were carried out manually using Microsoft Word 365.

## 3. Results

### 3.1. Sample Characteristics

Forty-nine RTs participated (i.e., 30 physiotherapists and 19 occupational therapists) in one-time sessions (four focus group sessions, three dyadic interviews, thirty-one one-on-one interviews). Sessions ranged between 23 and 85 min. The average length for focus groups, dyadic, and one-on-one interviews was 74, 69, and 44 min, respectively. The RTs represented seven provinces in Canada, with most identifying as female (n = 38), married (n = 29), and working in an urban location (n = 40). Participant characteristics are displayed in Table 1.

### 3.2. Identified Themes

The four overarching themes developed using thematic analysis were (1) navigating uncertainty along with ever-changing practices, policies, and attitudes, (2) morphing roles within a constrained system, (3) witnessing patients suffering and experiencing moral distress, and (4) recognizing the personal toll of the pandemic on self and others, and the lessons learned from the pandemic (see Table 2).

#### 3.2.1. Navigating Uncertainty along with Ever-Changing Practices, Policies, and Attitudes

The first theme is about how the RTs in this study identified an initial stressor of COVID-19 as “uncertainty” or the “fear of the unknown” (PT001) about the virus itself, personal protective equipment (PPE) requirements, and the changing messaging from administrators. One RT described this experience in the following quote, “There was that period, at the start, when we didn’t have enough protective equipment, and things were changing so quickly. There were changes in the infection control practices where we were going backwards and forwards and backwards and forwards and could never make pretty much sense of it at times” (PT003). Others also described enacted policies as “silly” when, to the RT, they did not make logical sense, as noted in the following quote, “they just would make these policies without really thinking of the consequences or sort of the rational components of it.” (OT002). There was a sense of “chaos” or working in a “twilight zone” (P013) despite having access to information from administrators, according to some, which contributed to “information overload” (P012).

There were undertones of fear when recalling the beginning days of the pandemic. The fear seemed to manifest from questioning, “Are we going to be the next Italy?” (OT003), that “every single person will get it [COVID-19]” (PT007) and “everyone was going to die” (P014); however, RTs also reported a normalization process of first experiencing this fear response to caring for COVID-positive patients, and then being surprised by “how quickly we kind of fell into feeling somewhat normal to wearing a mask all day” (OT002). Few also mentioned that taking action helped to minimize their stress and have more control over the situation, “We were doing it for a cause, for the greater good. And so I think trying to focus on that was helpful to minimize the number of stressors that we had due to the uncertainties” (OT013).

Along with the normalizing process were reported changes, including the slowing down of policy change, perceived deficiencies in such policies improving or “getting corrected eventually” (PT015), and shifts in attitudes toward the virus and healthcare providers by the public. For example, one RT shared, “At the beginning of the pandemic, healthcare workers were heroes…beeping horns, free pizzas, and bagels. About a year later, eighteen months later, healthcare workers were the pariahs. We were enemy number one” (PT003). Several participants expressed some degree of “anger” and “frustration” dealing with individuals who may have disbelieved in the virus or the vaccines.

#### 3.2.2. Morphing Roles within a Constrained System

In response to the rapidly changing situation, this theme describes how RTs found their roles “morphing”, needing to prioritize an increasingly complex patient caseload in restricted care environments. The participants were redeployed to other clinical areas (e.g., intensive care units, long-term care homes) or COVID-19 assessment sites and performed swabbing or other administrative tasks. Only when there was a “lull” would they return to their original unit. In some cases, RTs were redeployed to perform very different roles (i.e., vaccine administrators, administrative) and had to “muddle through”. According to this RT, “My job changed significantly, so I was a nursing aid helping nurses for older people who were paralyzed. And I was holding people’s heads so they could change their bottoms. And I was helping change sheets” (PT016).
*Many RTs also shared experiences of making virtual adaptations to their delivery of patient care, including getting accustomed to “…new software, as well as maintaining confidentiality with client interactions” (PT010). While for some, virtual care seemed to work effectively, others questioned the quality noted in the following quote: “You’re reinventing something you’ve done for two decades. I can’t see or touch my patients, which I always thought was essential. I’m doing it through a computer and trying my best to help these patients navigate their injury” (PT009).*

RTs identified prioritizing complex caseloads as a significant stressor. They also described increasing pressure to discharge patients despite people being sicker and facing staffing shortages. The situation’s complexity is highlighted in the following quote, “I would say that the quality and the breadth and the depth of the care that we gave was not the same because our priorities shifted. Being in an acute care hospital, we focused on getting people out as quickly as possible, so we weren’t necessarily doing the same thorough, caring, thoughtful, patient job we typically do” (OT016).

According to some RTs, their workspace became restricted, which reduced their ability to perform certain activities with their patients, have access to their equipment, or be able to provide group interventions. Consequently, their workload increased significantly, “You couldn’t have patients in groups. You had to have them each in their group. So basically, we went from being able to see five people in maybe half an hour to 40 min to seeing each person individually in their room in isolation, which takes 20 min for each person” (PT013).

#### 3.2.3. Witnessing Patients Suffering and Experiencing Moral Distress

This theme describes another source of stress RTs expressed: the pandemic’s impact on their patients related to reduced therapy or delayed care and their resulting moral distress. From their perspective, “client care suffered” (PT001). RTs said that, in some cases, patients were stuck in their hospital rooms, receiving rehabilitation therapy only twice per week compared to twice daily, and as a result, according to several participants, they experienced a high degree of deconditioning, “I would say nine out of ten of my patients came out of these rehab hospitals worse than when they left the hospital” (PT002). Delay in elective surgery was another contributing factor to the deconditioning that RTs noticed, “…So some patients were coming in for a planned surgery that presented poor enough that you would think that it was not an elective procedure” (PT004). In the community, RTs were also aware of the resource issues for vulnerable populations, like those lacking access to virtual therapy technology.

Several RTs also observed how people’s cognition and mental health suffered from physical isolation, either from social distancing mandated by the government in the community or from the hospital visitor restrictions, “The social isolation just kind of tipped them over the edge, and we had lots of seniors coming into hospital delirious and because the Emergency Department was so busy and they can’t manage confused, agitated people, they were being sedated and physically restrained” (PT003).

Many RTs shared the despair, sadness, and moral distress they experienced in response to patients suffering and being unable to perform their role to full scope, “I carried this guilt that I had to abandon [pediatric clients] as if they didn’t matter. We all had to get deployed to the long-term care centers where everybody died. I did my best. I felt like no matter what I was doing, everyone was dying anyway” (OT014). Some shared their decision to seek professional help as a coping strategy for the psychological burden they experienced.

#### 3.2.4. Recognizing the Personal Toll of the Pandemic on Self and Others

In this theme, the personal burdens of the pandemic for the RTs were in the form of worries about avoiding and spreading COVID-19, insufficient access to PPE, feeling isolated from family and friends, and having financial stress related to reduced income or job instability. A significant stressor mentioned was the fear of contracting the virus and unknowingly exposing their patients or families to COVID-19. Compounding their fear about catching the virus was treating COVID-19-positive patients, “I remember seeing patients in their thirties, and you could tell they were very fit and healthy and on a ventilator. That kind of thing was crazy. And, just trying to make sure you are doing everything you can to keep yourself safe while also trying to help these people” (PT004). The fear and the guilt of potentially exposing their loved ones and patients to COVID-19 meant that the RTs limited their social networks, kept separate living quarters from their families, and missed important life events. “When I first got COVID, I couldn’t tell my parents, who were living abroad, and I struggled at work and personally from the illness. It was hard being single; there was nobody I could share it with. And on top of that, you weren’t allowed to interact with anybody in person either” (PT015). The loneliness and isolation RTs experienced was an added stressor and contributed to some hitting a “personal mental health low” (OT013).

Financial concerns were another source of personal stress for those who owned or worked in private therapy clinics. These RTs reported a loss of income or working at a reduced rate; however, even when restrictions eased, clinics continued to see fewer patients due to strict infection control practices, which meant higher costs for supplies yet still less revenue, “I think most of my stressors were more about being a business owner and having to lay people off and watching your business come as close to failure as you ever thought possible” (PT009).

### 3.3. Lessons Learned from the Pandemic

The RTs contributed insights into lessons learned from working through the pandemic (see Table 3).

#### 3.3.1. Needing Engagement with and Clearer Communication Strategies for RTs

According to several RTs, during the pandemic, leadership in healthcare organizations often delivered vague or seemingly impractical policies “with no connection with the frontline” (OT006). Consequently, participants suggested that managerial staff ought to be “honest and transparent” (OT013) with their staff about what is happening, even when there is uncertainty. Part of better communication is defining expectations for RTs to enable them to focus on delivering quality care, “This is the system that we’re going to work in, that we’re going highlight our priority. So, it’s very transparent. I want to do deep, good work where I can develop my skill set and provide consistently good care” (PT005).

#### 3.3.2. Addressing Workforce Issues through More Staffing and Funding

Another dominant lesson learned mentioned was staffing and ensuring sufficient coverage for illness and mental health breaks. Many also stated that increasing caseloads are not manageable and that more RTs are needed to tackle waitlists. “It needs funding, right? There needs to be more staff. In the system I was in before, the patients only got four visits. That’s not adequate rehab. I think a change in funding models can provide more service. So, whether that’s funding for more staff or more visits per staff” (PT014).

#### 3.3.3. Designing Innovative Models of Care

RTs suggested designing more innovative models of care. For example, some thought more virtual care could make RTs more efficient, “So if we were looking at trying to reduce wait times, being more efficient, why can’t we still keep virtual care…we can still do virtual visits for those straightforward assessments” (PT017). Another consideration was focusing on community-oriented rehabilitation programs to facilitate transitions from hospital to home. Some RTs saw health holistically and suggested: “more support services for mental health and addiction, more supportive housing for people with high care needs” (PT015).

## 4. Discussion

Many physicians and nursing staff experienced depressive symptoms and traumatic events during the pandemic [41,42]. Occupational differences among other disciplines have been mainly understudied. This study provides insight into the stressors experienced by Canadian registered RTs during COVID-19, which can mitigate future challenges of those practicing rehabilitation services and strengthen our pandemic preparedness.

Few studies have focused on RTs’ occupational and personal stressors during COVID-19 [43], and none qualitatively combine occupational therapists and physiotherapists. Our study offers a unique view of how RTs delivering rehabilitative services expressed significant concerns about *navigating uncertainty* while adapting their role and practice to changing policies, including redeployment to other clinical settings or COVID-19 testing sites. Redeployment to other teams and new tasks (e.g., virtual care) during the peak of the pandemic was experienced by many RTs, and according to the evidence, one-third of occupational therapists and physiotherapists were redeployed [44,45] and between 37.1% and 75% reported some or many changes in their service responsibilities [45,46]. However, over time, some participants seemed to normalize the abrupt changes. These findings support other studies’ results of a significant correlation between anxiety and resilience among healthcare workers [47,48]. Resilience is the healthcare worker’s ability to rise and adapt to challenging times [49], which may protect against mental health problems. A qualitative study captured the adaptive nature of occupational therapists’ profession to a more generic and supportive role across service than the traditional role [50]. Another source of stress was *morphing roles within a constrained system* or the rapid adoption of virtual care, which required RTs to adapt their professional conduct. Our findings suggested that virtual care provided a safe mode for patients to receive care. RTs either struggled with modifying their hands-on approach or considered virtual care an effective alternative. Overwhelmingly, virtual care was reportedly widely implemented as a mode of intervention delivery for rehabilitation services [51,52,53] with mixed responses [52,54]. Therefore, COVID-19 provided an opportunity to learn and potentially further develop models of virtual rehabilitation care [55], including adopting practical resources from established virtual rehabilitation programs [56].

It is undeniable that COVID-19 has impacted the mental health of RTs. According to the theme of *witnessing patients suffering and experiencing moral distress*, many participants reported moral distress resulting from patients suffering and personal worries for themselves and their loved ones. The morally stressful events transpired from increased workloads and insufficient resources, particularly in the context of rising infection rates, staffing shortages, and infection control practices. Moral distress examined by review studies demonstrated that during COVID-19, healthcare workers had to carry out tasks under extreme conditions that minimized their ability to influence the situation [57,58]. Consistent with our findings, other studies observed that healthcare workers caring for patients with COVID-19 showed significantly more moral distress, anxiety, and depressive symptoms than those not caring for COVID-19 patients [59], and in interviewed occupational therapists, moral distress was experienced as post-traumatic stress disorder [60]. Similar experiences of moral distress in rehabilitative professionals and other health professions during COVID-19 have been reported in the literature [61,62]. Witnessing patient harm in the form of reduced care, physical deconditioning, and social isolation while feeling helpless was also described in our study by RTs. These findings align with COVID-19 research of other healthcare providers observing neglect of patients’ care needs, failure to prioritize patient care, and feeling responsible for contributing to substandard care standards [62].

Participants expressed considerable fear of transmitting COVID-19 to their patients and their loved ones or becoming ill themselves from the virus in the theme of *recognizing the personal toll of the pandemic on self and others*. Compounding this unavoidable stress in our findings was the physical distancing restrictions and the inability to interact with family. The psychosocial impact of anxiety and distress was categorized as a natural response according to the WHO [63], given the rapidly and unpredictable changing conditions [64]. In this study, diminishing or inaccessible PPE supply among RTs contributed to their fear of being infected, infecting others or being isolated. Similarly, these findings have been observed in cross-sectional research that healthcare workers with insufficient/or no PPE were more likely to report depressive symptoms [65,66,67]. The psychological impact of the pandemic on frontline healthcare workers was profound [68], with high levels of stress, anxiety, depression, burnout and sleep disturbance in nurses and doctors [69,70,71]. Several prior studies reported a high incidence of depression, anxiety and insomnia in rehabilitation professionals [71,72], and more so in occupational therapists than in physiotherapists and nurses [73]. Experiences of job loss and reduced pay, especially from those in the private sector, resulted in additional psychological burdens for RTs. Higher mental health problems in healthcare workers have also been found in females [74], which may speak to other gendered roles, like parenting and household tasks. Prior studies also demonstrated the positive relationship between excessive financial hardship and worsening psychological stress [75], job performance [76], and suicidal thoughts and behaviors [77].

### Implications of Findings

The implications of our findings on the RTs needing clear communication support other evidence on protective factors [78]. Managing psychological distress could be accomplished by offering transparent and clear communication to minimize anxiety [79] and promote trust and a sense of control [80] among healthcare workers. Some RTs also recognized unreasonable policies that seemingly excluded input from frontline workers. This finding aligns with international evidence on communication and information strategies implemented in hospital settings. “Bottom–up” communication channels were absent, which, when making decisions, limited listening to and considering staff’s experience and expertise [81]. While some promising information and communication practices from hospital case studies have been identified [81,82], attention has also been paid to communication approaches supporting COVID-19 vaccination of the healthcare workforce [83,84] and the public [85]. Therefore, future comparative analyses could help identify the best health crisis communication practices.

Much work is needed to bolster the rehabilitation services workforce given the high need for rehabilitation for COVID-19 survivors [86] and the limited resources available [87]. Occupational therapists [88] and physiotherapists [89] are well-positioned to manage the full range of consequences; however, the pandemic’s tidal wave of new challenges may require innovative rehabilitation approaches to care. For example, telerehabilitation therapy programs could offer quick and effective options instead of traditional, in-person visits where clinically appropriate. Virtually delivered rehabilitation has proven effective in certain clinical populations [90,91]. Work is already underway in Canada with the Provincial Post-COVID-19 Rehabilitation Response Framework (PCRF) to promote three care pathways across the care continuum for individuals affected by post-COVID-19 conditions [92]. The PCRF provides a framework that providers and health systems could adapt to local communities to reduce unmet rehabilitation needs. However, more research is needed to understand the implementation and sustainability factors of novel rehabilitation interventions supported by RTs.

Our study offers several strengths. It is the first to study the combination of Canadian occupational therapists and physiotherapists’ COVID-19 experiences and stressors. Our robust sample size and combination of one-on-one interviews, dyadic interviews, and focus groups provided rich, contextual data across different practice settings within seven provinces. The noted limitations include the sampling technique and the lack of demographic data about racial and ethnic diversity. Knowing that demographic variables interact in various ways, conducting an intersectionality analysis could have informed the findings. The purposeful sampling technique was applied without considering maximum variation. As a result, the authors cannot confirm the sample’s data representativeness.

## 5. Conclusions

Due to the rapid transmission of the virus and its immense impact on human loss, the healthcare workforce was profoundly affected. Our study demonstrated the experience of RTs in response to COVID-19, including their navigating uncertainty, adapting their roles in response, and bearing witness to patient suffering, including patient isolation and functional decline. Consequently, they suffered moral distress, poor mental health, and some from challenging financial situations, especially those in the private sector. However, many expressed a resilient attitude in response to these stressors. The implications learned going forward include identifying promising communication strategies that could act as protective factors addressing workforce constraints and diminishing resources through innovative models of care.

## Figures and Tables

**Table 1 healthcare-12-01050-t001:** Characteristics of participants.

	Physiotherapists (n = 30)	Occupational Therapists (n = 19)
**Age (years)**		
**25–29**	2	1
**30–34**	4	5
**35–39**	7	6
**40–44**	6	2
**45–50**	5	2
**50+**	4	3
**Unknown**	2	-
**Gender**		
**Female**	22	16
**Male**	8	3
**Marital status**		
**Married/common-law**	18	15
**Single**	9	4
**Divorced**	3	-
**Location**		
**Urban**	25	15
**Rural**	1	
**Mix**	4	1
**Setting**		
**Hospital**	17	12
**Community**	6	7
**Private practice**	7	-
**Years in practice**		
**0–5**	4	1
**6–9**	5	6
**10–14**	4	4
**15–19**	8	-
**20+**	10	9

**Table 2 healthcare-12-01050-t002:** Summary of Findings.

No.	Theme	Focused Coding	Illustrative Quote
Section 3.2.1	*Navigating uncertainty along with ever-changing practices, policies, and attitudes*	- Trying to make sense of COVID-19 practices - Sensing palpable fear - Trying to keep self and others safe	*“If something needed to be done, she [Critical Care supervisor] would get it done and she would get it done by the end of the day like she was just a superstar. But I think in some ways that also backfired for people like me when it was information overload, and it wasn’t so much a trigger, but it was just like, Oh my God, another I can’t keep it all straight and not that it triggered. But it was just too much.” (PT012)*
Section 3.2.2	*Morphing roles within a constrained system*	- Covering different units - Being tasked with seeing ICU patients - Working with more acutely ill patients	*“For a couple of weeks I covered the ICU. That was definitely super interesting but super nerve wracking experience um cause we had a lot of patients who were coming from other hospitals because they didn’t have enough space and I was kind of tasked with and having to try to decide, you know, who was well enough to see and a lot of them were actually too sick for me to even do much physio treatment with.”* *(* *PT004);*
Section 3.2.3	*Witnessing patients suffering and experiencing moral distress*	- Patients bouncing back and forth - Patients receiving less physical therapy - Patients in worse state	*“It’s just that moral distress of people not getting the care they need and watching our case as a leader, watching our caseloads grow, like watching our wait list grow um when something we’d battle to kind of bring down. That was really hard to watch” (OT011).*
Section 3.2.4	*Recognizing the personal toll of the pandemic on self and others*	- Being weary of symptomatic patients - Trying to acquire PPE - Having financial stress	*I think most of my stressors were more about being a business owner and having to do things that I never would have anticipated having to do, like laying people off and watching your watching your business come as close to failure as you ever thought possible (PT009).*

**Table 3 healthcare-12-01050-t003:** Summary of Lessons Learned.

No.	Theme	Focused Coding	Illustrative Quote
Section 3.3.1	*Needing engagement with and clearer communication strategies for RTs*	- Interprofessional communication still a disconnect - Listening to frontline workers more	*“I think that’s partly because we weren’t at the table helping make those decisions. So I think it needs to be we need to be at the table, but we’re not.” (PT012)*
Section 3.3.2	*Addressing workforce issues through more staffing and funding*	- More funding for allied health professionals, and equipment - Incentivizing new graduates - Pay considerations	*“Distribute the workload a little bit more evenly uh and try and work on staff retention um so a little bit more like focus on activities on team building. Um focus a little bit more on, you know, the motivation for you to come to work versus making it a chore.” (OT008)*
Section 3.3.3	*Designing innovative models of care*	- Using family caregivers more - Family mobilizing patients - Leveraging virtual care to increase access	*“And it’s even not just hospital staff, but like community care for example, is so underfunded that if you had funded these people, they wouldn’t have come to you guys in the first place.” (PT015)*

## Data Availability

The original contributions presented in the study are included in the article. Further inquiries can be directed to the corresponding author.

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
