# Peer review of "From Crisis to Opportunity: A Qualitative Study on Rehabilitation Therapists’ Experiences and Post-Pandemic Perspectives"

_healthcare, 2024, doi:10.3390/healthcare12101050_

Round 1

Reviewer 1 Report

Comments and Suggestions for Authors

I congratulate you on your novel and important study, about understudied professionals who suffered various problems during the COVID-19 pandemic.

Some suggestions and comments:

1) Your study seems to have a high risk of intrinsic representativeness bias, as there were only 49 self-selected volunteers (even though they were from the 7 provinces). Can you better detail the selection and representativeness of the sample? Could you explain it or notify it among the “Limitations of the study”.

2) It would be interesting if they had added a quantitative data table, with percentages of the clinical characteristics or other problems suffered, or proposed it for future studies, in the limitations and/or conclusions.

3) It says on line 402: “Our strong sample size.” It may be a fairly small sample size. I propose replacing the phrase with: “Despite a limited sample size, the combination of one-on-one interviews…”

4) On line 413, it says: "and some challenging from financial situations" As a CONCLUSION, it seems not very specific detailing the findings of the study. I propose to add the phrase: ", especially from those in the private sector".

Author Response

Please see letter.

Reviewer 2 Report

Comments and Suggestions for Authors

Thank you for the opportunity to review the paper "From Crisis to Opportunity: A Qualitative Study on Rehabilitation Therapists’ Experiences and Post-Pandemic Perspectives"

It's an interesting topic in terms of mental health, and very pertinent. Congratulations on the topic.

I have a few suggestions for improving your paper.

Abstract

In the abstract, the conclusion does not respond to the aim of the study.

You must not repeat the keywords with the words that are already in the title of the article. Choose others.

Introduction

What is the relevance of developing this study in occupational therapists and physiotherapists? It's not clear from the introduction, you should develop this further.

Materials and Methods 

Were the interviews conducted virtually recorded? If yes, they should record it and say so, if no, they should also mention this.

Results

Why didn't you use a table to put the themes/questions that were asked, in terms of perception, formatting and clarity, it's much more simple. Reading points 3.2 and 3.3 would be much more relevant and interesting if the information was organized in a table.

Discussion

The discussion does not really discuss the data found in the study, there is no discussion of the findings of the content of the interviews. They do not discuss the data in table 1, nor do they understand why table 1 is necessary. They should revise this section.

Conclusion

They still don't answer the proposed objective.

Author Response

Please see letter. 

Round 2

Reviewer 2 Report

Comments and Suggestions for Authors

Thanks again for the opportunity to review the paper.

I would still like to suggest the inclusion of a small table in points 3.2 and 3.2, as this is a qualitative study!

As in the link that shows Healthcare:https://www.mdpi.com/2227-9032/11/14/2013, it is possible to see a simple table, with organised information that helps reading to be clearer and less fatiguing.

Author Response

Dear Reviewer two,

Thank you for your suggestion about the inclusion of tables for the findings. We have addressed your comment by adding two summary tables.